# Assessment of transcriptomic constraint-based methods for central carbon flux inference

**Siddharth Bhadra-Lobo**[1]*, **Min Kyung Kim**[1], **Desmond S. Lun**[1,2,3]

**1** Center for Computational and Integrative Biology, Rutgers, The State University of New Jersey, Camden, NJ, United States of America, **2** Department of Computer Science, Rutgers, The State University of New Jersey, Camden, NJ, United States of America, **3** Department of Plant Biology, Rutgers, The State University of New Jersey, New Brunswick, NJ, United States of America

* sid.bl@rutgers.edu

**Data Availability Statement:** All relevant data are within the manuscript and its Supporting Information files.

**Funding:** This work was supported in part by NSF award no. 1515511 (MKK).

## Abstract

### Motivation

Determining intracellular metabolic flux through isotope labeling techniques such as $^{13}$C metabolic flux analysis ($^{13}$C-MFA) incurs significant cost and effort. Previous studies have shown transcriptomic data coupled with constraint-based metabolic modeling can determine intracellular fluxes that correlate highly with $^{13}$C-MFA measured fluxes and can achieve higher accuracy than constraint-based metabolic modeling alone. These studies, however, used validation data limited to *E. coli* and *S. cerevisiae* grown on glucose, with significantly similar flux distribution for central metabolism. It is unclear whether those results apply to more diverse metabolisms, and therefore further, extensive validation is needed.

### Results

In this paper, we formed a dataset of transcriptomic data coupled with corresponding $^{13}$C-MFA flux data for 21 experimental conditions in different unicellular organisms grown on varying carbon substrates and conditions. Three computational flux-balance analysis (FBA) methods were comparatively assessed. The results show when uptake rates of carbon sources and key metabolites are known, transcriptomic data provides no significant advantage over constraint-based metabolic modeling (average correlation coefficients, transcriptomic E-Flux2 0.725 and SPOT 0.650 vs non-transcriptomic pFBA 0.768). When uptake rates are unknown, however, predictions obtained utilizing transcriptomic data are generally good and significantly better than those obtained using constraint-based metabolic modeling alone (E-Flux2 0.385 and SPOT 0.583 vs pFBA 0.237). Thus, transcriptomic data coupled with constraint-based metabolic modeling is a promising method to obtain intracellular flux estimates in microorganisms, particularly in cases where uptake rates of key metabolites cannot be easily determined, such as for growth in complex media or *in vivo* conditions.

**Competing interests:** The authors have declared
that no competing interests exist.

## Introduction

Computational tools integrating transcriptomic data into genome-scale metabolic models can predict system-level and condition specific metabolic flux distributions. Many methods for inferring metabolic fluxes from gene expression data have been, and continue to be, developed [1–3]. However, the comparative performance of these methods lacks diverse experimental flux data for validation. Existing validation was performed exclusively against flux data generated from *E. coli* and *S. cerevisiae* (yeast) cultures grown on glucose as the sole carbon source [3, 4]. Cells cultured on identical substrates utilize highly similar metabolic pathways [5]. This carbon source bias presents significant similarities in the measured metabolic flux distribution across previous validation datasets which may have been inadequate in assessing predictive performance.

Carbon source availability and relative uptake rates influence cellular metabolism. In nature, heterotrophic microorganisms can encounter a wide set of possible carbon sources to support growth, including sugars, polyols, alcohols, organic acids, and amino acids [6]. Heterotrophs such as *E. coli* and *Bacillus subtilis* have been widely studied and cultured on a variety of substrates including monosaccharides (e.g. glucose, fructose, galactose), disaccharides (e.g. sucrose), and two-carbon compounds (e.g. acetate) [7–11]. Thus, under a multitude of possible carbon sources, an incorrectly constrained heterotrophic model can reduce the predictive accuracy of central carbon fluxes from conventional FBA methods. Gene expression may be useful to impute model constraints based on transcript abundance in the absence of specific carbon source and uptake rate data.

Growth condition encompasses the availability of metabolic state-determining metabolites, both organic and inorganic (e.g. glucose, $CO_2$, photons, $NO_3$). Missing or incorrect growth condition information can change flux predictions to alternate metabolic states of the cell. Photoautotrophic unicellular metabolic models are generally well characterized and therefore simpler to constrain with respect to carbon source. The depletion of non-carbon metabolites may metabolically adapt the cell to alternate metabolic states. For example, light inhibition can shift metabolism from either autotrophic, heterotrophic, or a combination of both as mixotrophic in *Synechocystis* sp. PCC 6803 [12]. A substrate void of nitrate can induce replenishing of nitrogen from metabolic sinks such as amino acids for *Synechococcus* sp. PCC 7002 [13]. In the lack of environmental condition specificity, informational deficit may be overcome with gene expression data such as key pathways being allocated flux values based on the upregulation of associated transcripts.

Previous studies [2, 3] have extensively evaluated the predictive capability of *in silico* flux prediction using measured extracellular and intracellular fluxes in multiple experimental conditions, but under single carbon source bias (glucose) in two organisms. To address the limitations of the previous dataset, we have compiled an additional 21 experimental conditions of transcriptome measurements coupled with corresponding central carbon metabolic intracellular $^{13}$C flux measurements in 4 organisms (8 in *E. coli*, 8 in *Bacillus subtilis*, 3 in *Synechocystis* sp. PCC 6803, and 2 in *Synechococcus* sp. PCC 7002). These conditions were applied to models run using two transcriptomic methods (E-Flux2 and SPOT) [4] and the non-transcriptomic method parsimonious FBA (pFBA) [14]. E-Flux2 and SPOT were chosen as representative transcriptomic methods, and similarly pFBA was chosen as the representative non-transcriptomic method, because prior publications suggest they are among the best in their respective method classes [3, 4]. In this study, the generality of E-Flux2 and SPOT have been validated against pFBA using this new dataset of diverse carbon sources and conditional constraints.

In the absence of carbon source and growth condition data, transcriptomic coupled constraint-based modeling is useful in bridging this information gap. If it is even feasible in the

experimental condition of interest, the extraction of $^{13}$C-labeled isotopes is costly and laborious. Additionally, many published $^{13}$C-MFA studies are unable to be reproduced due to missing information, with "only 30% of the studies examined were found to be acceptable" in a review by Crown et al. [15]. The $^{13}$C-labeled data also conveys minimal growth condition information as it cannot be directly applied to non-carbon metabolites [16]. In contrast, gene expression data is relatively simple to gather and is obtained from cell culturing experiments regularly. With transcriptomic FBA methods, researchers can utilize their gathered expression data to estimate intracellular metabolism.

## Materials and methods

### Gene expression, flux datasets, and metabolic models

All gene expression measurements obtained were not normalized any further past the instrument processed signal. Any log-transformed data was transformed back to their original scale by exponentiation.

**Data and model for *E. coli*.** For *E. coli*, both the measured gene expression (single color microarray) and $^{13}$C flux data were obtained from a previous study by Gerosa et al. [17]. In this study, data were measured from *E. coli* wild type BW25113 cells growing exponentially on eight different carbon sources: glucose, galactose, gluconate, fructose, glycerol, pyruvate, acetate, and succinate. We used iJO1366 [18] as the genome-scale metabolic model.

**Data and model for *B. subtilis*.** For *B. subtilis*, we used transcriptomic (single color microarray) and $^{13}$C flux data published in [19] and [20], respectively. Data were obtained from *B. subtilis* BSB168 cells grown under eight conditions defined by different carbon sources: glucose, fructose, gluconate, succinate + glutamate, glycerol, malate, malate + glucose, and pyruvate. For the genome-scale metabolic model of *B. subtilis*, the model published by Oh et al. [21] was used.

**Data and model for *Synechocystis* sp. PCC 6803.** For *Synechocystis* sp. PCC 6803, transcriptomic (RNA-seq) data was graciously provided by Dr. Le You (University of California San Diego, USA) and Dr. Yinjie Tang (Washington University in St. Louis, USA) [12]. The $^{13}$C flux data was compiled from three different publications [12, 22, 23]. Data were measured from the strain *Synechocystis* sp. PCC 6803 grown under three different conditions: photoautotrophic (i.e. $HCO_3^-$ (bicarbonate) as the main carbon source) [23], photomixotrophic (i.e. open air $CO_2$ + glucose) [22], and heterotrophic (i.e. open air $CO_2$ + glucose, constrained photons) [12], respectively. We used the genome-scale metabolic model of *Synechocystis* sp. PCC 6803 developed by Knoop et al. [24]. An external pseudo-compartment was added to the model through which metabolites can be exchanged with the external environment via cellular transport reactions.

**Data and model for *Synechococcus* sp. PCC 7002.** For *Synechococcus* sp. PCC 7002, the transcriptomic (RNA-seq) data was obtained from a previous publication by Ludwig and Bryant [25]. The $^{13}$C flux data for this model was gathered from Qian et al. [26]. Data were measured from *Synechococcus* sp. PCC 7002 cells grown photoautotrophically (i.e. $CO_2$ carbon source and photon uptake) with 10 mM nitrate and with no other nitrogen source. iSyp821 was used for the organism's genome scale-metabolic model [13].

### Computational prediction and correlation

**Computational metabolic flux prediction.** In this study, E-Flux2, SPOT [4], and pFBA [14] were used to predict metabolic flux distributions. Biomass production was set as the objective function for E-Flux2 and pFBA. All FBA methods used in this study are referenced from their original publications [4, 14]. Computations were carried out on the macOS Mojave

platform using a personal computer with a 3.1 GHz Intel Core i5 processor with 8GB of RAM. E-Flux2, SPOT and pFBA methods are implemented in MOST (Metabolic Optimization and Simulation Tool) which is available at http://most.ccib.rutgers.edu [27].

**Correlation calculations.** Validation of the predictive accuracy of the methods used in this study was done by calculating the uncentered Pearson product-moment correlation between *in silico* fluxes and corresponding $^{13}$C determined intracellular fluxes as previously described in [4]. A value of the correlation coefficient close to +1 or -1 indicates a strong relationship via a positive or negative scale factor, respectively, between experimentally measured fluxes and computationally predicted fluxes; a value of 0 indicates no such relationship [28]. If a measured reaction corresponds to a set of consecutive reactions in the model that are linked with intermediate metabolites (AND relationship), then the minimum flux value among the predicted fluxes was used. If a measured flux corresponds to multiple identical reactions (OR relationship), the sum of those predicted fluxes was used to calculate the correlation.

Correlations were calculated between the measured and predicted fluxes per carbon source in MATLAB R2018b (The Math Works Inc., Natick, Mass., USA). The predicted fluxes for the transcriptomic methods (E-Flux2 and SPOT) were generated using the respective carbon source and/or growth condition gene expression profile. pFBA does not use gene expression and was run in two scenarios, one where the carbon source flux was not specified (i.e. maximal uptake allowed) and one where the carbon source flux is specified (for uptake rates used see S1 Table in S1 File). Carbon source fluxes were gathered from uptake rates from the respective $^{13}$C flux experiments (mmol/g DCW/h).

## Results and discussion

To test generality of E-Flux2 and SPOT, we evaluated predictive accuracy by calculating the uncentered Pearson correlation (Section Methods) between experimentally measured and computationally predicted intracellular fluxes using transcriptomic data, for the compiled 21 experimental conditions. The dataset consists of 8, 8, 3, and 2 conditions of *E. coli*, *B. subtilis*, *Synechocystis* sp. PCC 6803, and *Synechococcus* sp. PCC 7002, respectively (Section Methods provides carbon source information). We expect model choice affects the transcriptional FBA methods more than the non-transcriptional. A less complete model may have reduced constraint mapping from the relevant gene expression data.

We have chosen the uncentered Pearson correlation as a good, goodness-of-fit metric because transcriptomic flux inference, in general, estimates that high transcript count corresponds with high flux, but not the actual flux value. Therefore, the predicted flux values are in arbitrary units. This type of correlation captures predictive accuracy irrespective of the scaling introduced by the gene expression data.

Testing flux prediction under known and unknown carbon sources, *E. coli* and *B. subtilis* fluxes were simulated under different carbon source availabilities, at three different stages labeled as AC, DC, and Full AC.

- DC: Known carbon source and uptake rate information available, uptake rate is only supplied to the non-transcriptomic method, pFBA.

- AC: Unknown carbon source and uptake rate, only eight speculative carbon sources without uptake rate data are available to the model.

- Full AC: No carbon source information available, all possible carbon sources (and any other extracellular metabolites) opened for exchange into the model.

Testing flux prediction under different growth condition in PCC 6803 and PCC 7002. Fluxes were simulated based on the organism's possible metabolic states, at two different stages. Carbon sources are fewer and simpler to constrain in these photoautotrophic organisms, therefore here AC is the same as Full AC in the previous heterotrophic organisms.

- DC: Growth condition information and metabolic state are known and uniquely applied to simulate each respective organism's metabolic states. Carbon uptake rate data only supplied to pFBA.

- AC: No growth condition information is available, all possible carbon and inorganic metabolites available for simulating the mixotrophic condition.

Unknown growth condition was used to demonstrate cases of complex media or *in vivo* growth of cultures. An example of this would be in studying the metabolism of enteric bacteria, in which the growth medium is complex, and the culture is grown *in vivo*. An example enteric model is *Mycobacterium tuberculosis*, in which using conventional [13]C-MFA to measure latent bacteria would not be feasible due to additional constraints such as tissue specificity and slow *in vivo* growth rates, but extraction of RNA expression data has been shown to be possible [29, 30]. In the cases of gut microbiome, the distribution of bacterial species in the gut has been shown to vary based on diet [31]. With improved RNA extraction techniques, it may be possible to detect microbial metabolic shifts in species that continue to persist in the gut during dietary changes, using transcriptomic flux prediction. A conjectured experiment would be to sample RNA from the gut during a period of one type of host diet, then sample RNA again after a period of time on another diet. Although this is highly dependent on the quality of expression profiles and metabolic models.

## Correlations for known and unknown carbon source

**Central carbon flux correlations in *E. coli*.** Under direct carbon source (DC) the *E.coli* models were supplied with only one carbon source each (Fig 1A). With complete carbon source information supplied, correlation between the transcriptomic and non-transcriptomic methods are similarly good. pFBA was provided an additional constraint to improve prediction with the experimentally measured carbon source uptake flux (uptake rate) being set within the pFBA runs only (Fig 1A). For a speculative set of possible carbon sources, Fig 1B shows the measured fluxes of *E. coli* grown on a single carbon source correlated with the predicted fluxes when supplied with all 8 carbon sources (AC) used in the measurements (i.e. glucose, galactose, gluconate, fructose, glycerol, pyruvate, acetate, and succinate) per model. Fig 1C simulates the absence of any carbon source and uptake rate information, with the model fully open for exchange with the extracellular environment. Here overall predictive accuracy drops across all methods as the number of available carbon sources increases. E-Flux2 on average performs comparably to SPOT, with slightly worse correlation on average. Models run with all 294 available carbon sources (Full AC) and 30 ion sources, shows that on average E-Flux2 and SPOT generate reasonable correlations (Fig 1C). All three methods produce lower correlations for carbon sources found in the TCA cycle (Fig 1C Full AC acetate, pyruvate, and succinate). These low correlations were investigated and determined to be due to predicting flux opposite in direction to the measured flux (S3 Fig in S1 File). The measured fluxes for glycolysis are negative in reaction direction and the predictions are positive, while the measured fluxes for TCA cycle reactions are positive, and the predicted fluxes are negative. SPOT maintains higher correlations compared to E-Flux2 and pFBA due to predicting the TCA cycle reactions in the correct direction.

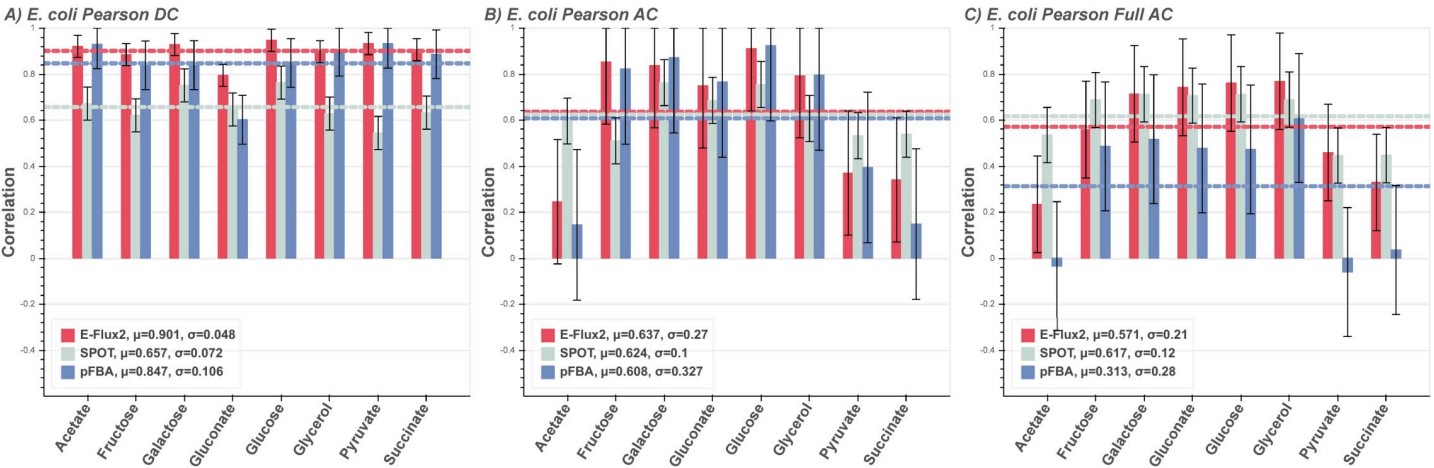

**Fig 1. *E. coli* predictions.** Correlations between measured and predicted flux of *E. coli* grown on 8 different carbon sources for the three FBA methods E-Flux2 (red), SPOT (gray), pFBA (blue). Horizontal dashed lines are the respectively colored mean correlations per method. The mean ($\mu$) is the average prediction correlation per method. The standard deviation ($\sigma$) is the spread of prediction correlation above and below the mean, denoted by the error bars. **(A)** Respective direct carbon source (DC) supplied. pFBA was given the additional constraint of known uptake rate in the single carbon source, while E-Flux2 and SPOT were not. All methods perform consistently across the individual carbon sources. **(B)** All 8 carbon sources supplied and correlated with measured flux from single carbon growth (AC). Correlations drop in all methods, particularly in the TCA cycle carbon sources (Acetate, Pyruvate, and Succinate). **(C)** All possible carbon sources in the model supplied (Full AC). All methods again lose performance, but the transcriptomic methods retain reasonable correlations. See Supplementary **S1 Table** for uptake rates used.

**Central carbon flux correlations in *B. subtilis*.** The *B. subtilis* measured fluxes consist of 8 different carbon sources, with two cases of double carbon sources experiments (Fig 2 glutamate + succinate and malate + glucose). Fig 2A shows the DC correlations from E-Flux2 and pFBA is comparable, with known carbon flux giving the best correlations on average. In speculative carbon sources, Fig 2B, all three methods perform similarly on average for AC. pFBA performs similarly poorly to the other methods for the double carbon cases and only

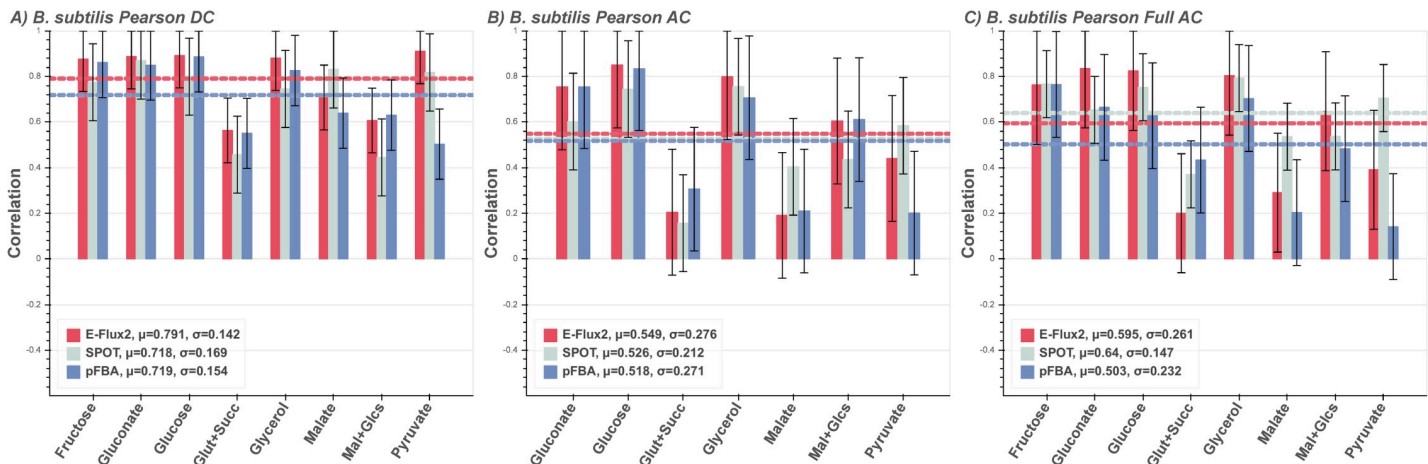

**Fig 2. *B. subtilis* predictions.** Correlations between measured and predicted flux of heterotrophic *B. subtilis* grown on 8 different carbon sources for the three FBA methods E-Flux2 (red), SPOT (gray), pFBA (blue). Double carbon sources are glutamate plus succinate (Glut + Succ) and malate plus glucose (Mal + Glcs). Horizontal dashed lines are the respectively colored mean correlations per method. The mean ($\mu$) is the average prediction correlation per method. The standard deviation ($\sigma$) is the spread of prediction correlation above and below the mean, denoted by the error bars. **(A)** Respective direct carbon source (DC) supplied. pFBA was given the additional constraint of known uptake rate in the single carbon source, while E-Flux2 and SPOT were not. All methods perform consistently across the individual carbon sources, with minor drops in correlation for double carbon sources (Glut + Succ and Mal + Glcs). **(B)** All 8 carbon sources supplied and correlated with measured flux from single carbon growth (AC). Correlations drop in all methods, particularly for Malate. **(C)** All possible carbon sources in the model supplied (Full AC). All methods again lose performance, but the transcriptomic methods retain reasonable correlations. See Supplementary **S1 Table** for uptake rates used.

marginally better for glutamate + succinate (see Discussion). SPOT performs the best for the TCA cycle single carbon source cases (AC malate, AC pyruvate). The same can be seen in the Full AC models (269 carbon sources, 25 ion sources) (Fig 2C) but pFBA on average performs worse, most notably in the TCA cycle single carbon sources.

## Discussion of known and unknown carbon source

For the *E. coli* and *B. subtilis* models, if carbon source and uptake rates are known, the directly provided carbon source and uptake rate information (DC) produces flux predictions in non-transcriptomic pFBA that are comparable to transcriptomic E-Flux2 (Figs 1A and 2A). SPOT provides a reasonable, but lower average prediction for both DC cases. E-Flux2 predicts flux similarly to pFBA except E-Flux2 was not provided any uptake rate information. The effect of gene expression derived reaction bounds predicts central carbon flux well, even without providing respective carbon uptake rates. This suggests that gene expression can serve as a substitute for measured carbon source uptake information, if the carbon source is known.

If carbon source is speculatively known, uptake rate is unknown, and presented with a relatively small set of 8 possible carbon sources (AC), pFBA predictive power drops significantly (Fig 2A and 2B). Without transcriptomic data, pFBA sets fixed proportion uptake rates of the available metabolites in the model across multiple cases. This affects the subsequent central carbon flux prediction as a single flux pattern is being predicted across all conditions. In contrast, the transcriptome coupled methods do not have the same uptake of carbon source per condition, as the gene expression dictates the proportions of carbon source flux for cellular uptake. This suggests that with unknown uptake rates and speculatively known carbon sources, gene expression can still serve as a substitute for measured uptake rate data.

Under both unknown carbon source and unknown uptake rates (Full AC), where the models are allowed uptake of all possible carbon sources present in the model, the pFBA average prediction score drops further while E-Flux2 and SPOT remains similar to their AC correlations (Figs 1C and 2C). The E-Flux2 and SPOT average correlation even increases slightly from the *B. subtilis* AC to Full AC cases. A possible explanation is that in the overabundance of carbon sources, the gene expression can mediate the allocation of flux feeding into central carbon metabolism when presenting from multiple metabolic network entry points and thereby predict reaction directionality better (see S3–S5 Figs in S1 File). This is in contrast to when flux directionality is set based on a small set of carbon sources, such as the TCA cycle or glycolysis relevant metabolites.

Additionally, in the Full AC model, all ion uptake reactions were open, suggesting the transcriptome can also facilitate ion flux prediction, where $^{13}$C data generally does not provide information. For both unknown carbon source and unknown uptake rate conditions (Figs 1C and 2C), SPOT performs the best on average. This is likely due to SPOT maximizing correlation with flux prediction and the gene expression set, rather than setting expression-based reaction bounds (as in E-Flux2) which can set a large flux window that can affect predicted directionality in subsequent reactions (see S3–S5 Figs in S1 File). The generally higher prediction correlations for E-Flux2 and SPOT suggest that under both unknown carbon source and unknown uptake rates, gene expression data can substitute for carbon source and uptake rate information for central carbon flux prediction.

On average the transcriptomic methods perform better than pFBA under unknown carbon source and uptake, but in one exception of the double carbon source conditions, pFBA predicts central carbon flux with higher accuracy than either transcriptomic method across the DC, AC, and Full AC cases (Fig 2A, 2B and 2C glutamate + succinate). This is potentially due to pFBA predicting low flux correctly for a subset of the measured flux values for *B. subtilis*,

while E-Flux2 and SPOT allocated different fluxes for these reactions based on the presence of the associated transcripts (see reaction directions in S5 Fig in S1 File). Hence, when a measured reaction has low flux, but some transcript abundance, the transcriptomic methods may attribute more flux to these reactions.

Additionally, carbon source similarity affects flux predictions. On a carbon source basis, glutamate + succinate measured fluxes are similar to glycerol and pyruvate measured fluxes. The other double carbon source (malate + glucose) exhibits a measured flux distribution very close to the single carbon malate measured distribution (see S1A, S5 Figs in S1 File). This suggests that some carbon sources produce similar flux distributions to others, both experimentally and *in silico*. This is supported by the clustering of pFBA flux patterns across all constraints and conditions (S1B, S5 Figs in S1 File) which shows similarity between the predicted overall glutamate + succinate distribution to glycerol and pyruvate predicted distributions. This effect has also shown to shift flux predictions away from the measured distribution. In one case, the predicted distribution for malate + glucose more closely resembles the predicted glucose distribution, but in the measured flux patterns the malate + glucose measured flux distributions more closely resembles the malate flux distribution.

## Correlations for known and unknown growth condition

**Central carbon flux correlations in PCC 6803.** In *Synechocystis* sp. PCC 6803 autotrophic, mixotrophic, and heterotrophic conditions (Fig 3A) E-Flux2 and pFBA produce very similar central carbon flux distributions under the autotrophic condition. These predictions correlate well with the autotrophic measured fluxes, suggesting that both methods are producing nearly identical flux distributions. In the mixotrophic condition, pFBA, with known carbon source and flux, produces a higher correlation than the other methods. All methods predict heterotrophic central carbon metabolism poorly, with SPOT predicting the only positive correlation between measured and predicted fluxes. SPOT produces similar correlation values with the three measured flux distributions, and the only non-negative correlation consistently for all three conditions. Fig 3B shows the correlations of fluxes predicted using the three conditional gene expression sets (expression data collected from autotrophic, mixotrophic, and heterotrophic cultures) while under mixotrophic constraints, simulating how predicted fluxes correlate under unknown conditions and guided by transcriptomic data. E-Flux2 and pFBA produce negative correlations for all mixotrophically constrained predictions. SPOT again provides the only positive correlations.

**Central carbon flux correlations in PCC 7002.** Measured fluxes from *Synechococcus sp.* PCC 7002 in nitrogen-replete (10 mM nitrate) and nitrogen-deprived (N-deprived, no nitrogen source) conditions correlated well with predicted fluxes under autotrophic constraints. SPOT produced significantly better central carbon flux for the N-deprived condition and the other methods performed similarly across both nitrogen conditions (Fig 4, N-replete). Fig 4B shows PCC 7002 in an AC mixotrophic condition not naturally exhibited in PCC 7002 (see Results and discussion, Unknown carbon source and growth condition). Both sets of predicted fluxes are allowed open uptake of all carbon sources as well as $NO_3$ uptake, simulating unknown carbon source and unknown nitrate availability. SPOT performs well under the set of unknown conditions, while the other methods perform poorly.

## Discussion of known and unknown growth condition

For the cyanobacteria models (PCC 6803 and PCC 7002), carbon source is relatively easy to choose and constrain. The models we assessed are known to fix a single source of inorganic carbon under autotrophic condition, which pFBA can predict well with known carbon source

**A) PCC 6803 Pearson DC All Conditions**

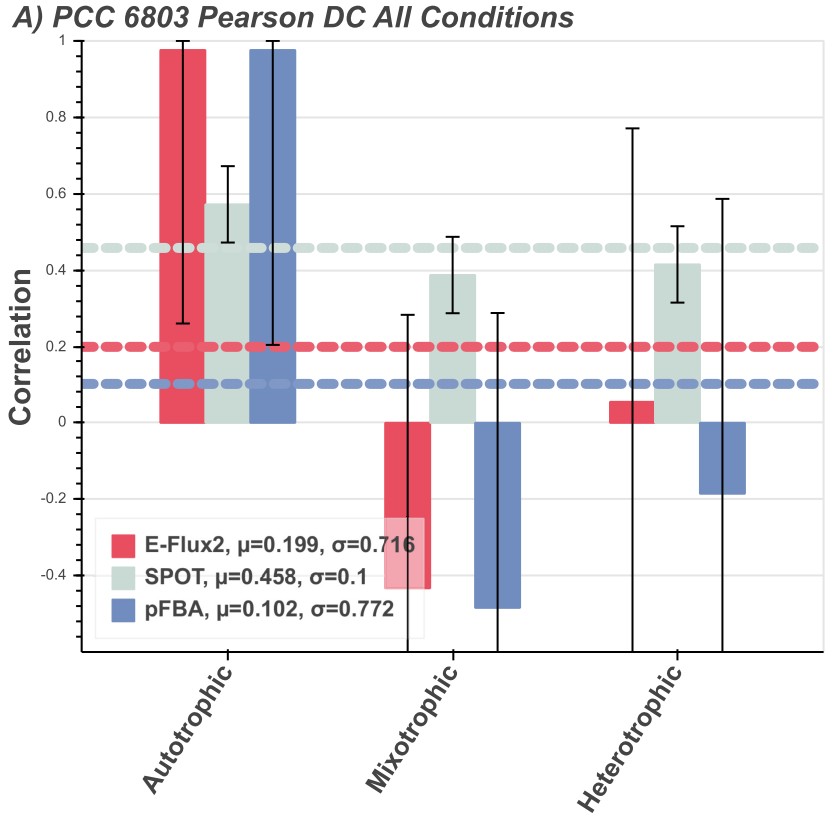

**B) PCC 6803 Pearson AC Mixotrophic Condition**

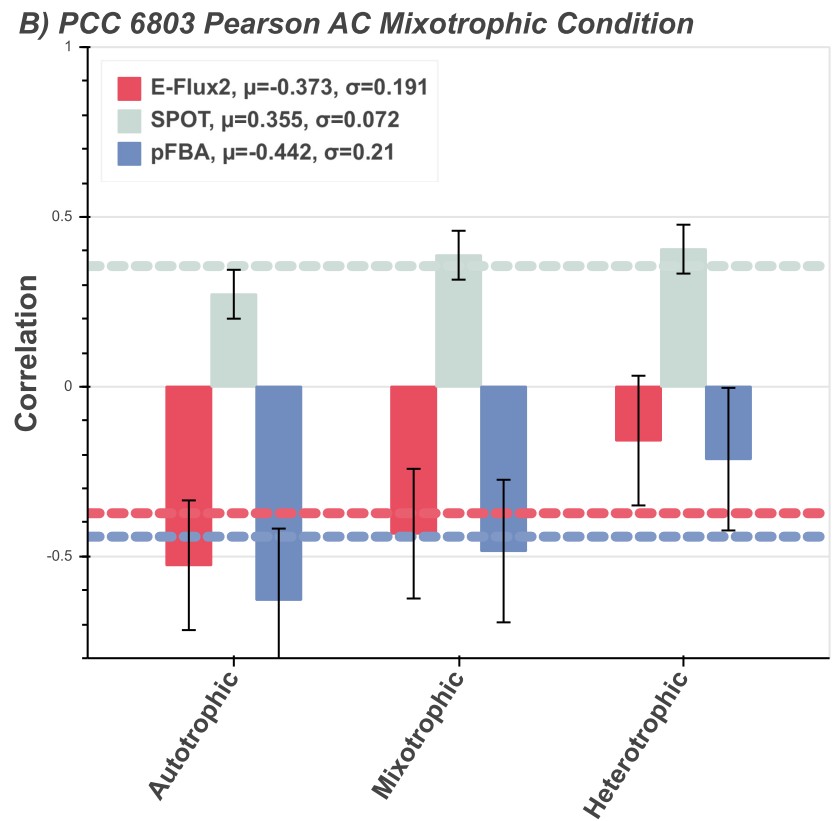

**Fig 3. *Synechocystis* sp. PCC 6803 predictions.** Correlations between measured and predicted flux of multitrophic *Synechocystis sp*. PCC 6803 grown in 3 different environment conditions for the three FBA methods E-Flux2 (red), SPOT (gray), pFBA (blue). Horizontal dashed lines are the respectively colored mean μ, correlations per method. The mean (μ) is the average prediction correlation per method. The above and below the mean. A) Autotrophic, mixotrophic, and heterotrophic conditional constraints standard deviation (σ) is the spread of prediction correlation applied and correlated with respective measured fluxes, denoted by the error bars. pFBA was given the additional constraint of known uptake rate in the single carbon source, while E-Flux2 and SPOT were not. B) Mixotrophic condition constraints applied and correlated with all three conditional measured fluxes. See Section Methods for condition specific model constraints and see Supplementary **S1 Table** for uptake rates used. for uptake rates used.

and uptake rates (Fig 3A). In the autotrophic growth condition, uptake rate of the inorganic carbon source does not significantly affect central carbon flux prediction (see S2 Fig in S1 File). But if an organism can increase biomass in multiple possible growth conditions (PCC 6803) then information pertaining to the presence and uptakes rates of inorganic carbon source versus glucose is much more useful.

With unknown growth condition information for PCC 6803, a reasonable approach for modeling the flux distribution is under mixotrophic conditions. That is, allow uptake of both inorganic and organic carbon as well as photon flux and use the associated gene expression to dictate how fluxes should be allocated. Fig 3B shows that under such conditions, pFBA and E-Flux2 predict similarly poor central carbon flux. However, SPOT consistently produces positive correlations between the predicted and measured fluxes, across growth conditions. This suggests that with SPOT, gene expression can give some idea of what the condition an organism is growing under using gathered gene expression and the genome-scale metabolic model. A possible explanation for the lower predictive accuracy in both E-Flux2 and pFBA compared to SPOT, is that under glucose availability the typical glycolysis flux distribution is not always found it nature (S6 Fig in S1 File). In PCC 6803, we found fluxes in the pentose phosphate pathway (see S2 Table in S1 File), which is an alternative metabolic route to glycolysis, has significantly higher flux predicted through it for SPOT in comparison to the other methods. This is further supported by information suggesting that PCC 6803 is merely a facultative heterotroph and therefore only metabolizes exogenous organic carbon when given no other choice [32].

In PCC 7002, the growth condition is only partially known. PCC 7002 is modeled under photoautotrophic conditions, but key secondary metabolite uptake rates are unknown ($NO_3$ exchange). Here pFBA predicts central carbon flux poorly. By applying different uptake rates of non-carbon metabolites, it is possible to determine whether an organism is in one metabolic state versus another. For example, constraining the uptake of oxygen can produce a flux distribution for anaerobic metabolism [33]. Similarly, in PCC 7002 the presence and depletion of nitrate to the system can lead to different intracellular carbon utilization.

PCC 7002 is known to be an obligate photoautotroph [34]. Therefore, non-transcriptomic methods should be able to perform well in predicting central carbon metabolism, but Fig 4 shows that pFBA given known carbon source and uptake rate performs worse than the transcriptomic methods in both N-replete and N-deprived cases. In Fig 4A SPOT predicts N-deprived central carbon flux better than the other methods. This likely due to the drawing of flux from the nitrogen sinks such as amino acids in order to accommodate for the lack of extracellular nitrate.

## Extended analysis: Artificial conditional information deficit

As an extension of our findings, PCC 7002 was constrained under a second artificial growth condition set to mimic mixotrophic conditions. We attempted to predict flux using the second condition's set of incorrect conditional constraints and see how gene expression might help

### A) PCC 7002 Pearson DC Autotrophic

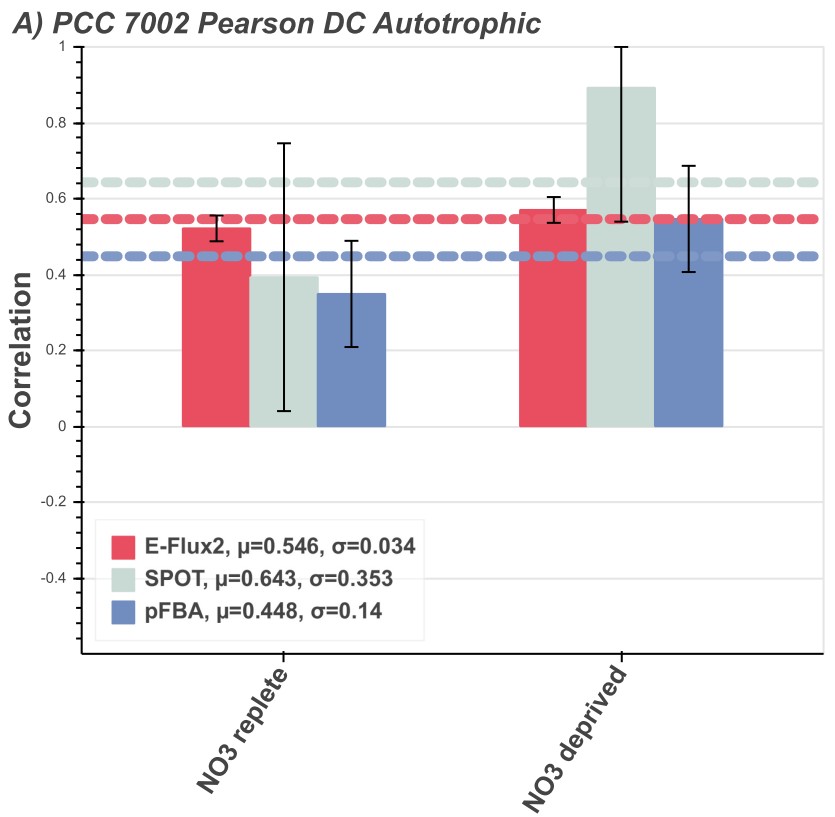

### B) PCC 7002 Pearson AC Mixotrophic

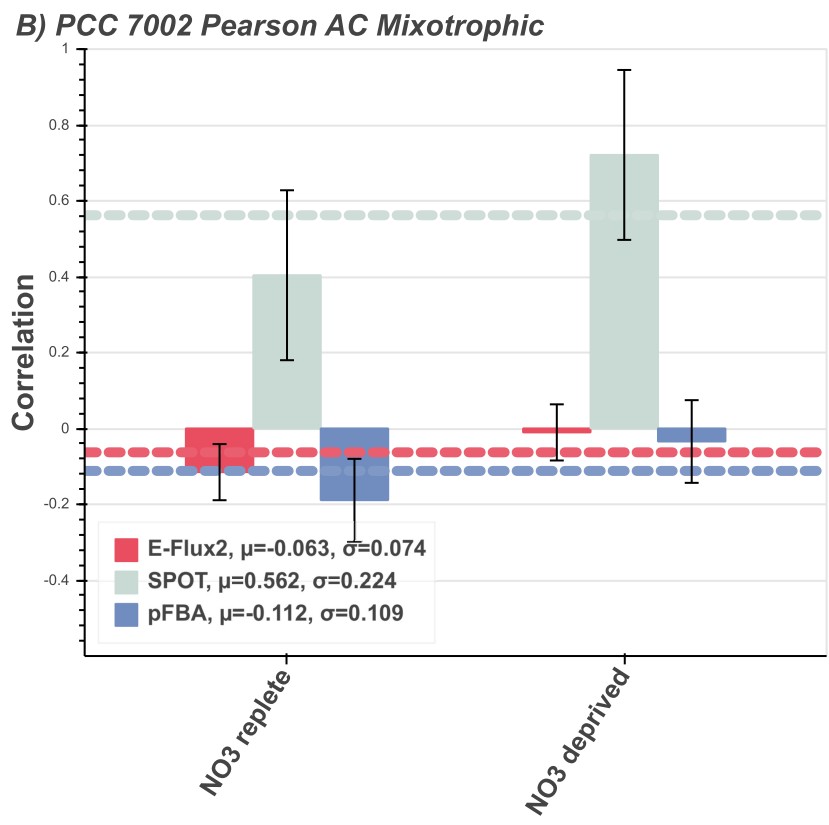

**Fig 4. *Synechococcus* sp. PCC 7002 predictions.** Correlations between measured and predicted flux of multitrophic *Synechococcus sp*. PCC 7002 grown in autotrophic conditions for the three FBA methods E-Flux2 (red), SPOT (gray), pFBA (blue). Horizontal dashed lines are the respectively colored mean correlations per method. The mean (μ) is the average prediction correlation per method. The standard deviation (σ) is the spread of prediction correlation above and below the mean, denoted by the error bars. **(A)** Autotrophic conditional constraints applied and correlated with N-replete and N-deprived measured fluxes. Supplementary Materials for uptake rates used. **(B)** PCC 7002 in AC autotrophic condition (mixotrophic, see Discussion) and unconstrained NO3 uptake. See Section Methods for condition specific model constraints and see Supplementary **S1 Table** for uptake rates used. for uptake rates used.

reduce prediction error. This allows for carbon sources other than $CO_2$ allowed for uptake as well as unconstrained $NO_3$ uptake for both the N-replete and N-deprived cases ([Fig 4B]). This mixotrophic state is not found in nature, and therefore the PCC 7002 central carbon flux distribution correlation was expected to be poor [34]. With the nitrate growth condition unspecified in the model, $NO_3$ was allowed into the cell freely for both conditions. The correlations for E-Flux2 and pFBA were indeed poor, but SPOT produced strong correlations. This suggests that even in incorrectly constrained models supplied with unrealistic carbon sources and no secondary metabolite information, gene expression can still be used to predict central carbon flux well (S7 Fig in [S1 File]).

## Conclusion

In this study, we compiled 21 experimental conditions and corresponding transcriptomic data for cells grown on various carbon sources and conditions. The predicted fluxes were correlated against experimentally measured fluxes to evaluate the predictive power of E-Flux2 and SPOT compared with the non-transcriptomic method, pFBA. pFBA is a representative method for comparison as it was shown to have good predictions, was used in the previous two validations studies, and does not use transcriptomic data [2, 3].

If carbon source and uptake rate information are accurately known for microorganisms and gene expression data is unavailable, pFBA is a suitable method for central carbon flux prediction (Figs [1A] and [2A]). Even with well-defined carbon source, uptake rate, and growth condition information (other factors on the cell's metabolism, such as light intensity), E-Flux2 performed better than pFBA in 13 of the 21 models. In all of these cases E-Flux2 was not provided any measured uptake rate data, while pFBA was.

If a carbon source or growth condition informational deficit is encountered, then SPOT is the method of choice as it consistently produced good correlations and can account for noncanonical internal metabolism (see Results and discussion, Known and unknown growth condition). Although pFBA can give good predictions, any uncertainty in carbon source or growth condition carries the risk of generating very poor predictions. Even with accurate carbon source and growth condition information pFBA can still produce negative correlations ([Fig 3A]). Gene expression can produce better central carbon flux as the expression data can account for other unknowns in the model, beyond just the carbon source ([Fig 4A], N-deprived).

Based on the findings in this study, we propose a general decision tree to be used in constraint-based modeling for central carbon flux prediction in microorganisms ([Fig 5]). In this figure, if no expression data is available, then pFBA is the default method of choice. If any expression data is available, then a transcriptomic method is suggested as gene expression has been shown to account for additional informational deficits beyond carbon source such as ion exchanges.

Using validated methods like SPOT can minimize the risk of predicting incorrect central carbon flux distributions in the absence of accurate carbon source and growth condition data.

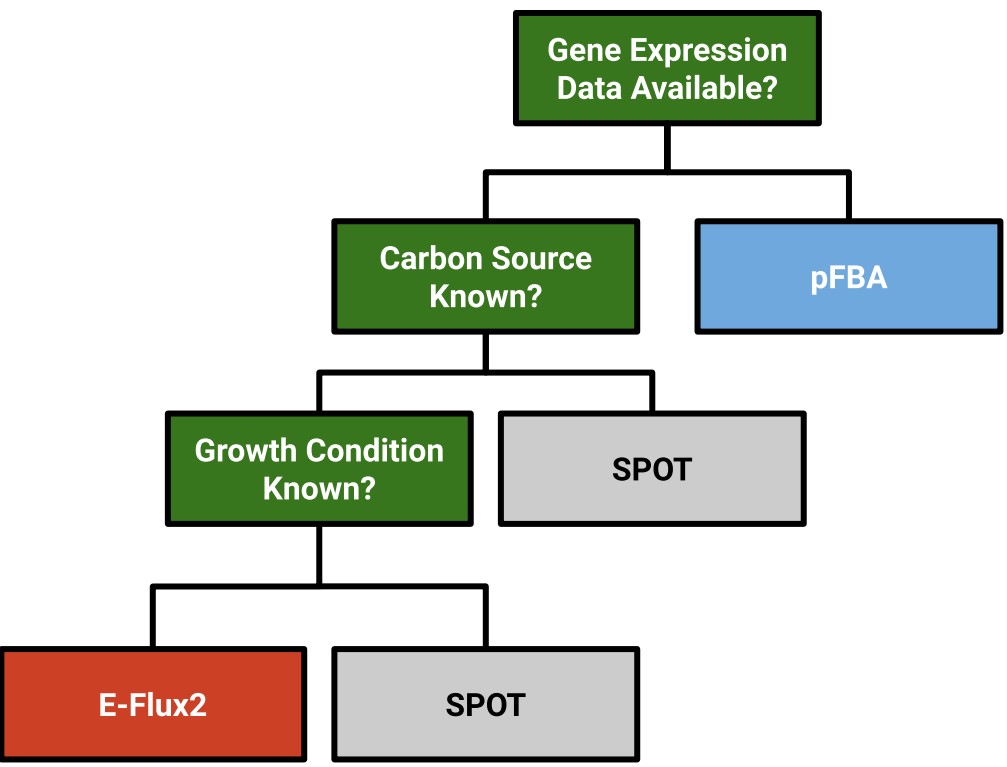

**Fig 5. General methods decision tree.** The three FBA methods are shown as E-Flux2 (red), SPOT (gray), pFBA (blue). Left branches on the tree indicate a YES decision, right branches indicate a NO decision. Growth condition refers to the availability of inorganic and organic metabolites that can shift metabolism between different states (e.g. photons s, $NO_3$, $CO_2$, glucose).

Not only does SPOT consistently produce positive correlations in all 21 samples, but also produces low if not the overall lowest, standard deviations in predictive accuracy (Figs 1–5, legend $\sigma$-values). For future improvement, developing a method for better determining flux directionality based on gene expression values should improve transcriptomic flux prediction.

In cells grown on well-defined media, it is relatively easy to determine carbon sources and uptake rates. The carbon source is generally known, while the uptake rate is determined from measuring how fast a culture consumes it. For cells grown *in vivo* or on complex media, where growth condition cannot be fully defined, $^{13}C$-labeling may not be even feasible, let alone the cost. Additionally, specifics pertaining to the growth conditions such as inorganic compound exchange may not be available or easily measured. In such cases, gene expression data can nevertheless be gathered simply and cheaply, and methods to infer intracellular metabolic flux from transcriptomic data (such as E-Flux2 and SPOT) have great utility.

## Supporting information

**S1 Dataset. *E. coli* model, data, and scripts used in this study.** The genome-scale metabolic model of *E. coli* iJO1366 [18]. Individual models in SBML format (.xml) with set constraints used AC, DC, etc. are included. Transcriptomic data (.csv) study by Gerosa et al. [17]. Predicted fluxes generated using these data. MATLAB scripts used for calculating correlations (.m).
(GZ)

**S2 Dataset.** *Bacillus subtilis* **model, data, and scripts used in this study.** The genome-scale metabolic model of *B. subtilis* from Oh et al. [21]. Individual models in SBML format (.xml) with set constraints used AC, DC, etc. are included. Transcriptomic data (.csv) from Oh et al. [19]. Predicted fluxes generated using these data. MATLAB scripts used for calculating correlations (.m).
(GZ)

**S3 Dataset.** *Synechocystis sp. PCC 6803* **model, data, and scripts used in this study.** The genome-scale metabolic model of PCC 6803 developed by Knoop et al. [24]. Individual models in SBML format (.xml) with set constraints used AC, DC, etc. are included. Transcriptomic data (.csv) from by Dr. Le You (University of California San Diego, USA) and Dr. Yinjie Tang (Washington University in St. Louis, USA) [12]. Predicted fluxes generated using these data. MATLAB scripts used for calculating correlations (.m).
(GZ)

**S4 Dataset.** *Synechococcus sp. PCC 7002* **model, data, and scripts used in this study.** The genome-scale metabolic model of PCC 7002 from Qian et al. [26]. Individual models in SBML format (.xml) with set constraints used AC, DC, etc. are included. Transcriptomic data (.csv) from Ludwig and Bryant [25]. Predicted fluxes generated using these data. MATLAB scripts used for calculating correlations (.m).
(GZ)

**S5 Dataset. Python code and tables used for correlations and plotting.** Python code used in plotting and analysis (.ipynb), and tab-delimited tables of correlations generated. See README for details.
(GZ)

**S1 File. Supporting figures and tables.** Supporting figures (S1–S8 Figs) and tables (S1 and S2 Tables) with associated captions.
(PDF)

## Acknowledgments

The authors thank James J. Kelley (Rutgers University, USA) for his work on Metabolic Optimization and Simulation Tool (MOST).

## Author Contributions

**Conceptualization:** Min Kyung Kim, Desmond S. Lun.

**Data curation:** Siddharth Bhadra-Lobo, Min Kyung Kim.

**Formal analysis:** Siddharth Bhadra-Lobo.

**Funding acquisition:** Min Kyung Kim, Desmond S. Lun.

**Investigation:** Siddharth Bhadra-Lobo, Desmond S. Lun.

**Methodology:** Siddharth Bhadra-Lobo, Min Kyung Kim, Desmond S. Lun.

**Project administration:** Desmond S. Lun.

**Resources:** Desmond S. Lun.

**Software:** Desmond S. Lun.

**Supervision:** Desmond S. Lun.

**Validation:** Siddharth Bhadra-Lobo, Desmond S. Lun.

**Visualization:** Siddharth Bhadra-Lobo.

**Writing – original draft:** Siddharth Bhadra-Lobo.

**Writing – review & editing:** Desmond S. Lun.

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
