## [Decision Letter · Decision Letter 0]

18 Oct 2019

PONE-D-19-22421

Assessment of transcriptomic constraint-based methods for central carbon flux inference

PLOS ONE

Dear Mr. Bhadra-Lobo,

Thank you for submitting your manuscript to PLOS ONE. After careful consideration, we feel that it has merit but does not fully meet PLOS ONE’s publication criteria as it currently stands. Therefore, we invite you to submit a revised version of the manuscript that addresses the points raised during the review process.

Please ensure all key points raised by the reviewers are addressed.  In particular both reviewers are concerned about variance and its impact on the comparison between experimental and simulation data.  Reviewer 1 highlights that the statistical analysis of the data is insufficient and that the data does not support some of the conclusion made (particularly as it relates to the performance of pFBA for the double carbon cases). These are a key concerns that must be clearly addressed in your revised manuscript.     

We would appreciate receiving your revised manuscript by Dec 02 2019 11:59PM. To enhance the reproducibility of your results, we recommend that if applicable you deposit your laboratory protocols in protocols.io, where a protocol can be assigned its own identifier (DOI) such that it can be cited independently in the future. For instructions see: http://journals.plos.org/plosone/s/submission-guidelines#loc-laboratory-protocols

We look forward to receiving your revised manuscript.

Kind regards,

Christopher N. Boddy, Ph.D.

Academic Editor

PLOS ONE

Journal Requirements:

Additional Editor Comments (if provided):

Reviewers' comments:

Reviewer's Responses to Questions

**Comments to the Author**

1. Is the manuscript technically sound, and do the data support the conclusions?

Reviewer #1: No

Reviewer #2: Yes

2. Has the statistical analysis been performed appropriately and rigorously? 

Reviewer #1: No

Reviewer #2: Yes

3. Have the authors made all data underlying the findings in their manuscript fully available?

Reviewer #1: Yes

Reviewer #2: Yes

4. Is the manuscript presented in an intelligible fashion and written in standard English?

Reviewer #1: Yes

Reviewer #2: Yes

5. Review Comments to the Author

Reviewer #1: Assessment of transcriptomic constraint-based methods for central carbon flux

inference

Summary: In this paper, the authors sought to understand which methods (E-Flux2, SPOT or pFBA) best predicts intercellular flux when certain data types for experiment is missing (e.g. uptake rates, primary carbon usage etc). By systematically comparing the results of the methods with experimental C-13 MFA, they were able to determine the power of each method to correctly predict intracellular metabolic flux given the data constraints. Furthermore, they also analyzed data from non-model organisms of genus Synechococcus to ensure that their assessment were not model organism specific. Based on these thorough comparisons, the authors propose a decision tree that helps modellers choose the best method given the available data (Figure 5).

While the authors were thorough in their data collection, they were not as thorough in their analysis. The paper can be greatly strengthened by presenting variance in correlations between the model prediction and the experimental result. Without this information, it is difficult to determine the significance of difference between the different methods. The paper could also be strengthened by benchmarking it against other algorithms for integrating transcriptomics data (e.g. GIMME and iMAT). Overall, the exhaustive data collected/ analyzed and the decision tree produced by the paper would be a very useful guide to modellers working with incomplete dataset.

Major Comments:

In cases of low correlations between predicted and experimentally measured fluxes, it would be interesting to know if any particular pathway or subsystem contributes disproportionately to low correlation. These may point to failure points in the model and direct future improvements. This is briefly addressed on Line 172, but an expanded analysis could strengthen the paper.

Linde 177-178: “In speculative carbon sources, Fig 2B, all three methods perform similarly on average for AC. pFBA performs the best for the double carbon cases..”

These conclusions are difficult to make for these conditions. pFBA’s performance on double carbon sources for AC is only slightly better (if at all). On ‘mal + glcs’ pFBA is on par with E-Flux2.

Figure 3B: Though the correlations are negative, there is a strong correlation between the pFBA flux prediction and the measured flux prediction (r = ~0.64). It is not clear from the text why the correlation is so strong and what this implies about the pFBA prediction in this sample/condition.

The crux of the paper relies on comparing correlations between experimental and simulation data. However, almost all correlations are presented as the mean with little information about variance. This makes it difficult to understand how significant the differences in correlations between the different methods are (see #3 above). We suggest the authors calculate total variance in correlation.

The authors should consider adding analysis with GIMME and iMAT. We understand the authors have previously done similar comparisons between GIMME, iMAT and E-Flux2 + SPOT (Kim et al. 2016). However, with the addition of new organisms in this paper, it would be of interest to know how these different methods perform relative to each other. If E-Flux2 and SPOT still outperform GIMME and iMAT in these new conditions, it may also lead to greater usage of their methods in the future.

Minor Comments:

If there are future rounds of reviews, please provide higher quality figures (specially figures 1 and 2). Currently they are difficult to read.

Line 165: Change “supplied with 8 the carbon sources” to “supplied with all 8 carbon sources.”

Figure 1B: The actual results should be described in the text beyond just description of what the analysis was (e.g. “Fluxes during glucose showed the highest correlation with….”).

Reviewer #2: This paper addresses a limitation of the subfield of metabolic network analysis concerned with predicting flux distributions under specific growth conditions (specified by key nutrient sources and/or maximum uptake rates). It collates 21 settings that have both Cabon-13 flux measurements (considered to be a gold standard here) as well as transcriptomic measurements that can be used to constrain a metabolic model. The models and the transcriptomic data are then used to predict fluxes, and these are then compared to the gold standard. The authors find that transcriptomic-based methods can be advantageous over standard methods when the transcriptomic data is obtained in a setting with incomplete information about the cell’s growth conditions.

Major comments:

1) I would have liked to see a more extensive discussion of the experimental error involved in measuring fluxes via Carbon-13; for instance, what is known about the repeatability of these measurements?

2) The correlation measure is very crude and hides a lot of nuances; for instance, not all fluxes are equally important to get right (the growth rate is typically the key quantity in model analysis, but also larger-scale pathway utilization may be of interest). Furthermore, it is very possible to have a perfect correlation, but be off by a factor of 10 in every single prediction - thus, not only should the correlation coefficient be reported, but also the equation of the line of best fit. An additional question could be about the similarities of the rankings between fluxes (e.g. Spearman correlation) because sometimes it is preferable to determine the relative importance of the reactions or pathways, rather than their absolute importance, to the cell’s internal state.

In addition, the use of the correlation as the main metric is not ideal because the typical goal of a metabolic study is to obtain qualitative information about the internal state of the cell (often relative to a baseline), whereas the correlation tries to make a quantitative statement about it, which is often (though not always) an over-interpretation of the metabolic model’s predictions.

3) It is unclear what the two growth conditions were used for PCC 7002 - only one of them is mentioned in the paper in the relevant section.

4) The figures (in particular, figures 1 and 2) are very grainy (low resolution) and do not use a consistent color scheme. They should be substantially improved. I recommend using ggplot2 in R for the plotting (alternatively, matplotlib in Python could work). You should also provide error bars on the plot, rather than having a mean and standard deviation specified in the legend.

5) The organization of the Results and Discussion section needs to be improved - the information is there, but hard to digest in the way it is currently organized. I would recommend having the sections represent an organism and the subsections represent the setting, or vice versa; right now it is a complicated mixture of both, which prevents an easy overview of the results.

6) I am not entirely convinced that the situation where the information about a cell’s growth conditions is incomplete is a realistic one. I would like to see the discussion at least one clear setting where that is likely to happen (or, barring that, an acknowledgment by the authors that the utility of the proposed approach is limited to a somewhat hypothetical scenario).

7) Can you learn something by combining the outputs of all three methods? For instance, is the average of the three predictions (or the two predictions by the methods using transcriptomics data) better correlated with the gold standard than the individual one? This could be an interesting finding in and of itself.

Minor comments:

Lines 43-44: Cells cultured on identical substrates produce highly conserved glucose metabolism pathways -> utilize highly similar pathways?

Line 66: To this end, we have […] -> specify to what end?

Line 83: Any log2 normalized data was exponentiated. -> I would say “any log-transformed data was transformed back to real scale by exponentiation” as normally, “exponentiated” suggests that the base of exponentiation was e, rather than 2.

Line 89: We used iJO1366 as the genome-scale metabolic model. -> I would have liked to see a discussion of the effect of the model choice on the results, though I recognize that this might be outside of the scope of the current article.

Line 101: remove the extra “H” before the formula for bicarbonate

Line 104: metabolites can exchange -> metabolites can be exchanged

Line 116: All methods used in this study are referenced with their original publications -> from their original publications? The sentence is very confusing in any case.

Line 118: When I follow the link I get the following error (please fix it): You don't have permission to access / on this server.

Line 139: see Methods 2.1 -> Section Methods 2.1

Line 310: incorrect hyphenation on both ends

6. PLOS authors have the option to publish the peer review history of their article (what does this mean?). If published, this will include your full peer review and any attached files.

Reviewer #1: No

Reviewer #2: No

---

## [Author Response · Author response to Decision Letter 0]

6 Dec 2019

Please refer to the attached file under the description "Response to Reviewers." and filename: Sid_PLosONE_rebuttal1_final.docx

We thank the reviewers for their time.

---

## [Decision Letter · Decision Letter 1]

22 Jan 2020

PONE-D-19-22421R1

Assessment of transcriptomic constraint-based methods for central carbon flux inference

PLOS ONE

Dear Mr. Bhadra-Lobo,

Thank you for submitting your manuscript to PLOS ONE. After careful consideration, we feel that it has merit but does not fully meet PLOS ONE’s publication criteria as it currently stands. Therefore, we invite you to submit a revised version of the manuscript that addresses the points raised during the review process.

Both reviewers agree that your revised manuscript addresses the major concerns from the first review.  They identify minor corrections that are needed for the manuscript to be acceptable. In particular, Reviewer 2's concern regarding M. tuberculosis must be addressed.  

We would appreciate receiving your revised manuscript by Mar 07 2020 11:59PM. To enhance the reproducibility of your results, we recommend that if applicable you deposit your laboratory protocols in protocols.io, where a protocol can be assigned its own identifier (DOI) such that it can be cited independently in the future. For instructions see: http://journals.plos.org/plosone/s/submission-guidelines#loc-laboratory-protocols

We look forward to receiving your revised manuscript.

Kind regards,

Christopher N. Boddy, Ph.D.

Academic Editor

PLOS ONE

Reviewers' comments:

Reviewer's Responses to Questions

**Comments to the Author**

1. If the authors have adequately addressed your comments raised in a previous round of review and you feel that this manuscript is now acceptable for publication, you may indicate that here to bypass the “Comments to the Author” section, enter your conflict of interest statement in the “Confidential to Editor” section, and submit your "Accept" recommendation.

Reviewer #2: (No Response)

Reviewer #3: All comments have been addressed

2. Is the manuscript technically sound, and do the data support the conclusions?

Reviewer #2: Yes

Reviewer #3: Yes

3. Has the statistical analysis been performed appropriately and rigorously? 

Reviewer #2: Yes

Reviewer #3: Yes

4. Have the authors made all data underlying the findings in their manuscript fully available?

Reviewer #2: Yes

Reviewer #3: Yes

5. Is the manuscript presented in an intelligible fashion and written in standard English?

Reviewer #2: Yes

Reviewer #3: Yes

6. Review Comments to the Author

Reviewer #2: I thank the authors for addressing my concerns; however, their modifications have introduced some additional concerns, as detailed below. I do not believe these to be a major obstacle to publication, however, and am therefore recommending minor revisions.

Major comments:

1) “enteric bacteria both pathogenic and commensal/mutualistic, in which the growth medium is complex,

and the culture is grown in vivo. An example of a pathogenic model is Mycobacterium tuberculosis.”

This incorrectly suggests that Mycobacterium tuberculosis is an enteric pathogen, which it is not (it is primarily, though not exclusively, found in the lungs)

In tuberculosis, the bacterium live inside of scar tissue of the lung and their metabolism is uncertain.

Technically, the bacteria (plural!) live inside a granuloma, formed by macrophages, rather than “scar tissue”.

Also, please provide references; in particular, the work by Palsson’s group on modeling Mycobacterium tuberculosis inside a macrophage seems relevant to cite (other models may have been published since as well).

“A hypothetical experiment would be to sample RNA from the gut during a period of one type of host diet, then sample RNA again after a period of time on another diet.”

Unfortunately, it is currently somewhat challenging to even determine the composition of a microbiome from RNA-Seq data at a particular timepoint with high accuracy; I think that expecting to detect changes in metabolism from this kind of data would be very ambitious (though some startups, such as Gusto, claim to be able to do that).

Grammar and minor comments:

simulated under based -> simulated based

Although this is dependent on the expression profiles and metabolic models to be complete enough for prediction of central carbon metabolism. - confusing!

decent correlations -> reasonable correlations

Double carbon sources are denoted as glutamate with succinate (Glut + Succ) and malate with glucose (Mal + Glcs). -> edit for grammar

If carbon source is speculatively known, and uptake rate is unknown, as presented with a relatively small set of 8 possible carbon sources (AC), pFBA predictive power drops significantly -> same

where more closely resembles -> same

S3 – S5 Fig -> Fig. S3 – S5 Fig

constrained under a second artificial growth condition was set to mimic -> edit for grammar

be used predict -> same

flux transcriptomic flux prediction. -> repeated word

S1, S2, and S3, S4, S5, S6, S7, and S8 Figs -> same comment as above

Reviewer #3: The authors present here the use of non-transcriptomic-based and transcriptomic-based methods to predict metabolic fluxes, where nutrient sources and uptake rates are available or not available. The authors show that non-transcriptomic-based and transcriptomic-based methods perform similarly with information available on carbon sources and uptake rates. However, transcriptomic-based methods are better at predicting flux distributions than constraint-based method, across various organisms and conditions.

The authors have sufficiently addressed the reviewer comments and the overall organization and structure of the manuscript has improved. I do not have further concerns except a few minor comments.

There is wording on nitrogen (or N) deprived throughout the manuscript. It should be nitrogen-deprived or N-deprived.

The terms DC, AC and full AC appear before their full names were introduced.

Line 253 effects -> affects

7. PLOS authors have the option to publish the peer review history of their article (what does this mean?). If published, this will include your full peer review and any attached files.

Reviewer #2: No

Reviewer #3: No

---

## [Author Response · Author response to Decision Letter 1]

5 Jul 2020

Responses are in the "Response to Reviewers" uploaded document. Thank you again for your time and consideration.

---

## [Editor Report · Decision Letter 2]

24 Aug 2020

Assessment of transcriptomic constraint-based methods for central carbon flux inference

PONE-D-19-22421R2

Dear Dr. Bhadra-Lobo,

We’re pleased to inform you that your manuscript has been judged scientifically suitable for publication and will be formally accepted for publication once it meets all outstanding technical requirements.

Kind regards,

Rajagopal Subramanyam

Academic Editor

PLOS ONE
---

## [Editor Report · Acceptance letter]

25 Aug 2020

PONE-D-19-22421R2 

Assessment of transcriptomic constraint-based methods for central carbon flux inference 

Dear Dr. Bhadra-Lobo:

I'm pleased to inform you that your manuscript has been deemed suitable for publication in PLOS ONE. Congratulations! Your manuscript is now with our production department. 

Kind regards, 

on behalf of

Prof. Rajagopal Subramanyam 

Academic Editor

PLOS ONE